# Branching mechanism of photoswitching in an Fe(II) polypyridyl complex explained by full singlet-triplet-quintet dynamics

Tamás Rozgonyi [1], György Vankó [1] & Mátyás Pápai [1✉]

It has long been known that irradiation with visible light converts Fe(II) polypyridines from their low-spin (singlet) to high-spin (quintet) state, yet mechanistic interpretation of the photorelaxation remains controversial. Herein, we simulate the full singlet-triplet-quintet dynamics of the $[Fe(terpy)_2]^{2+}$ (terpy = 2,2':6',2''-terpyridine) complex in full dimension, in order to clarify the complex photodynamics. Importantly, we report a branching mechanism involving two sequential processes: a dominant $^3MLCT{\rightarrow}^3MC(^3T_{2g}){\rightarrow}^3MC(^3T_{1g}){\rightarrow}^5MC$, and a minor $^3MLCT{\rightarrow}^3MC(^3T_{2g}){\rightarrow}^5MC$ component. (MLCT = metal-to-ligand charge transfer, MC = metal-centered). While the direct $^3MLCT{\rightarrow}^5MC$ mechanism is considered as a relevant alternative, we show that it could only be operative, and thus lead to competing pathways, in the absence of $^3MC$ states. The quintet state is populated on the sub-picosecond timescale involving non-exponential dynamics and coherent Fe-N breathing oscillations. The results are in agreement with the available time-resolved experimental data on Fe(II) polypyridines, and fully describe the photorelaxation dynamics.

[1] Wigner Research Centre for Physics, P.O. Box 49, H-1525 Budapest, Hungary. ✉email: papai.matyas@wigner.hu

Ultrafast experiments[1,2] utilising femtosecond optical and X-ray pulses have been extensively used to resolve the dynamics of excited-state molecules. However, the complexity of the time-resolved data and its analysis can be very high —especially in complicated cases such as transition-metal complexes—which often leads to ambiguities and contradictions in mechanistic interpretations. This is showcased by the light-induced low-spin→high-spin transition in Fe(II) polypyridyl complexes, which have attracted grand attention due to its potential importance in cutting-edge technologies, e.g., molecular data storage[3,4]. From femtosecond transient optical absorption[5] (TOAS), X-ray absorption[6,7] and X-ray emission spectroscopy[8,9] (XES) investigations, it is known that irradiation of the proto-typical $[Fe(bipy)_3]^{2+}$ complex (bipy = 2,2'-bipyridine) by visible light promotes the system from the low-spin ground state ($^1GS$) into singlet metal-to-ligand charge transfer states ($^1MLCT$), which in turn leads to conversion into the quintet high-spin state in <200 fs; this timescale is very similar for $[Fe(terpy)_2]^{2+}$ (terpy = 2,2':6',2"-terpyridine), another important member of the polypyridine family, as observed by TOAS[10]. On the other hand, it is still not clear according to which of the following two mechanisms is the high-spin state populated: i) a direct $^3MLCT→^5MC$ one, or ii) via a sequential $^{1,3}MLCT→^3MC→^5MC$ pathway with a triplet metal-centred intermediate ($^3MC$). While high time-resolution TOAS data[5] is claimed to support the direct mechanism, XES[8,9], owing to its sensitivity to the transition-metal spin state, clearly detects the involvement of a $^3MC$ state in the photorelaxation and thus indicates that the sequential pathway has to be operative. Furthermore, branching of the two mechanisms was proposed by a UV photoemission study[11].

Theory has a great potential to rationalise ultrafast experiments and provide complementary insights. However, the description of full singlet-triplet-quintet dynamics is extremely challenging due to the simultaneous treatment of disparate spin states, as well as multidimensional dynamics. For the description of the light-induced low-spin→high-spin dynamics in Fe(II) polypyridyl complexes, theoretical models have so far been limited to either the application of Fermi's golden rule[12–14], or quantum dynamics in exceedingly reduced dimensions[14,15]. While these studies do deliver valuable mechanistic insights, the limitations of the utilised theories (e.g., neglecting nuclear motion[12,13] or using less accurate electronic structure descriptions[15]) and large uncertainties in the calculated intersystem crossing (ISC) rates[13] do not allow to lift the controversies in the experimental observations.

Here, we simulate the full singlet-triplet-quintet dynamics of a Fe(II) polypyridine by full-dimensional spin-vibronic trajectory surface hopping. Importantly, we herein achieve the so far highest complexity in terms of nuclear dimensionality and electronic states, including all the $3N − 6$ nuclear coordinates as well as all accessible spin states in the simulation. We select the $[Fe(terpy)_2]^{2+}$ complex (Fig. 1a), the principal molecule of our investigations across the years[16–19], as a suitable represent of the Fe(II) polypyridines. Crucially, we present clear evidence that the sub-ps conversion to the quintet state occurs via the sequential pathway, with branching through the two $^3MC$ components, $^3T_{1g}$ and $^3T_{2g}$ (which differ in whether the $3d_{x^2−y^2}$ or $3d_{z^2}$ $e_g^*$ orbital is singly occupied); note that for simplicity, we use here octahedral notations. In addition, in agreement with a recent combined X-ray scattering/spectroscopy experiment[9], we observe non-exponential dynamics and coherent oscillations along the Fe-N breathing mode.

## Results and discussion

The diabatic potential energy curves along the Fe-N breathing mode, the most dominant normal mode of $[Fe(terpy)_2]^{2+}$ connecting the $^1GS$ (low-spin, LS) and $^5MC$ (high-spin, HS) states,

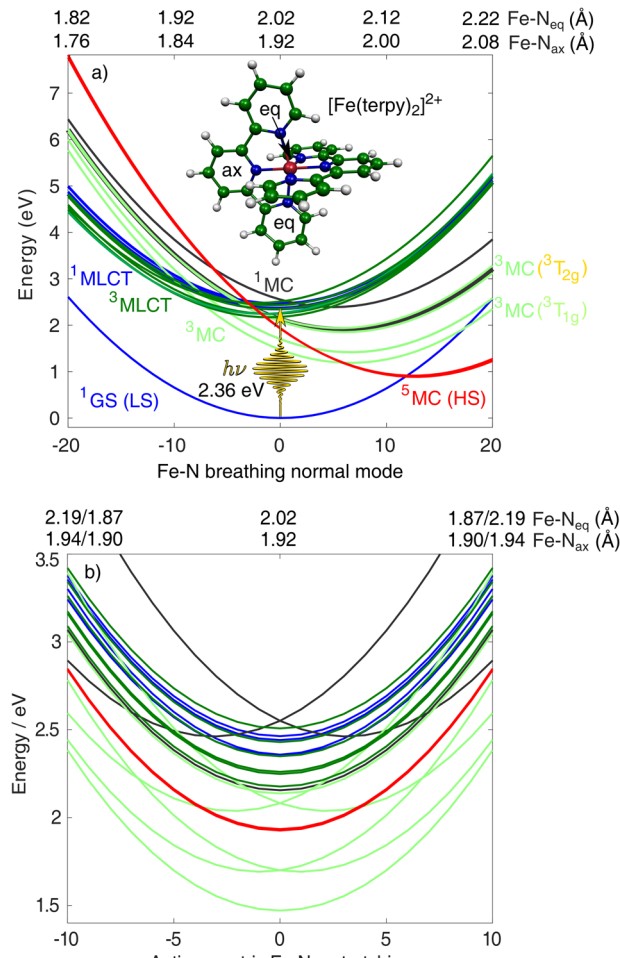

**Fig. 1 Diabatic potentials of $[Fe(terpy)_2]^{2+}$ along two normal modes.** The two panels show the potential energy surfaces along the **a** Fe-N breathing and **b** antisymmetric Fe-N$_{eq}$ stretching normal modes. The utilised normal mode coordinates are dimensionless (mass-frequency-scaled normal mode coordinates). Also shown is the molecular structure of the $[Fe(terpy)_2]^{2+}$ complex with "ax" and "eq" denoting the axial and equatorial nitrogen positions, respectively, as well as the $^1GS→^1MLCT$ excitation process ($\Delta E = 2.36$ eV).

are displayed in Fig. 1a. This represents a dimension along which the electronic transitions can be best interpreted, yet we need to take other modes into account that can couple the excited states. The potentials along one of such coupling modes, with anti-symmetric Fe-N bond stretching character, are displayed in Fig. 1b. The simulation produces relaxation trajectories on the potential energy hypersurface that involves all possible modes, and the presented results are obtained as average trajectories and populations from the numerous time-dependent runs. The excitation process occurs from $^1GS$ predominantly to the lowest-lying optically bright $^1MLCT$ states with a minor $^1MC$ component, as extracted from the simulated electronic populations at $t = 0$ fs. In Fig. 2a, we present the time-dependent diabatic excited-state populations as obtained by the full-dimensional dynamics simulations (solid lines; the utilised diabatisation procedure is described in Supplementary Note 1.). We note that the ground-state population is negligible for the present analysis, and is thus not shown here but discussed in the Supplementary Information, see Supplementary Note 1 and Supplementary Figure 1. From Fig. 2a, we observe a very fast (~50 fs) $^1MLCT→^3MLCT$ ISC, consistent with known photophysics of transition-metal

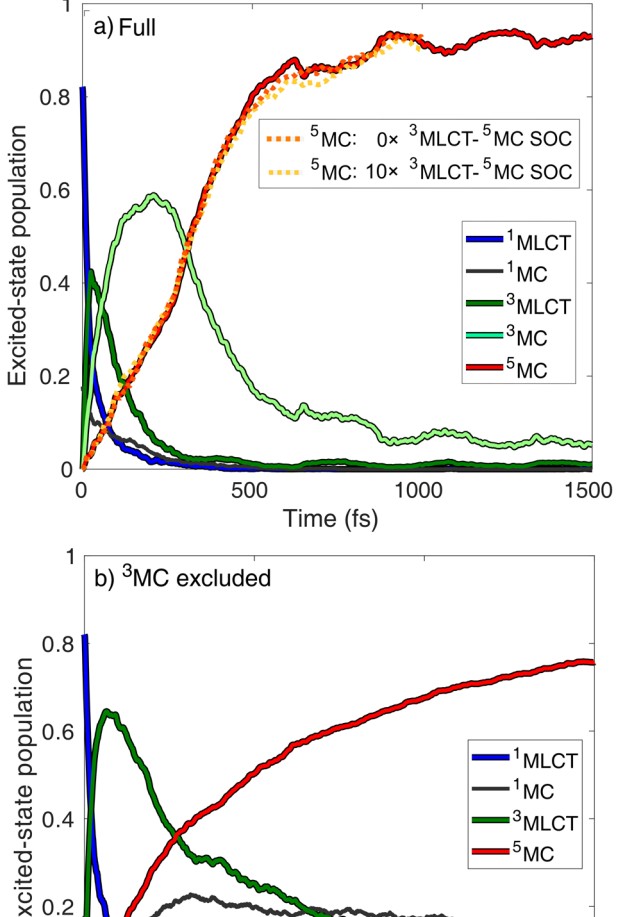

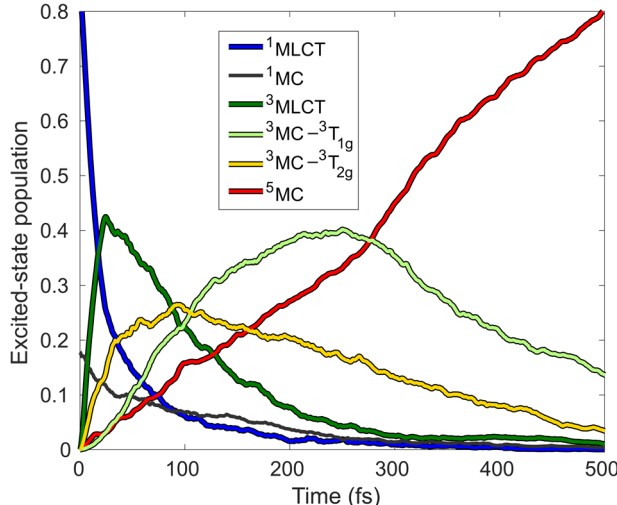

**Fig. 3 Simulated excited-state population dynamics of [Fe(terpy)$_2$]$^{2+}$ for early times $t \leq 500$ fs.** The $^3$MC population is decomposed to the $^3$T$_{1g}$ and $^3$T$_{2g}$ components.

**Fig. 2 Simulated excited-state population dynamics of [Fe(terpy)$_2$]$^{2+}$. a** Full simulation, the orange, and yellow dotted quintet population curves were obtained by decreasing the $^3$MLCT-$^5$MC SOCs to zero and enlarging them by 10×, respectively, while retaining all other parameters. **b** $^3$MC states excluded. The overall excited-state population is normalised to unity. The contribution of the ground-state population is small throughout the simulated dynamics (~10% and 0% at 1.5 ps for **a** and **b**, respectively), see Supplementary Figure 1.

complexes with singlet ground state[10,20–23]. Afterwards, the $^3$MC, and subsequently, the $^5$MC states are populated, to which latter state the conversion is nearly complete in ~500 fs. Importantly, the simulated dynamics shown in Fig. 2a (decay of MLCT states, participation of $^3$MC states, quintet population rise) are in very good overall agreement with the results of the most extensive time-resolved XES study on [Fe(bipy)$_3$]$^{2+}$,[9] which substantiates the accuracy of our results.

The quintet population rise (red curve in Fig. 2a) is clearly non-exponential, comprised of two apparent components, for ~0−250 fs and ~250−500 fs. Interestingly, we observe a very similar quintet population rise for the 0−250 fs region when excluding all $^3$MC states, as for the unrestrained simulation, see Fig. 2b; this seemingly assigns the 0−250 fs quintet component to the direct $^3$MLCT→$^5$MC transition (note that the quintet population here follows an exponential growth). However, we find that this direct $^3$MLCT→$^5$MC transition could only be operative in the absence of the $^3$MC states. This is revealed by

repeating the full simulation at varying $^3$MLCT-$^5$MC spin-orbit coupling (SOC) strength: the quintet population dynamics are found to be insensitive of decreasing the $^3$MLCT-$^5$MC SOC to zero, or increasing it by an order of magnitude, see the dotted lines in Fig. 2a. Importantly, this finding also shows that the sequential pathway via the $^3$MC states dominates the mechanism, even in the case of enhanced $^3$MLCT-$^5$MC coupling strength, due to e.g., nuclear distortions[13]. Therefore, we here conclude that the direct $^3$MLCT→$^5$MC process has a negligible role in the photorelaxation.

This result is rather stunning in the view of some of the so-far established mechanisms[5], and compelled us to explore further mechanistic intricacies of the photorelaxation, in particular, the involvement of $^3$MC states. In Fig. 3, we present more detailed excited-state population dynamics, unweaving the two $^3$MC components: the lower-energy $^3$T$_{1g}$ and the higher-energy $^3$T$_{2g}$, focusing on the timescale on which the ISC dynamics occur (0–500 fs). We can now identify that the quintet state is populated via a branching mechanism involving the two $^3$MC components: a faster $^3$MLCT→$^3$MC($^3$T$_{2g}$)→$^5$MC and a somewhat slower $^3$MLCT→$^3$MC($^3$T$_{2g}$)→$^3$MC($^3$T$_{1g}$)→$^5$MC process. Analysis of the diabatic populations along a representative set of individual trajectories confirms this mechanism with the $^3$MLCT→$^3$MC($^3$T$_{2g}$)→$^3$MC($^3$T$_{1g}$)→$^5$MC component being dominant (see Supplementary Note 2 and Supplementary Figures 3–5). We note that in principle there exists a third possible pathway, $^3$MLCT→$^3$MC($^3$T$_{1g}$)→$^5$MC, which we indeed observed (see Supplementary Figure 5) but its weight is so low that this channel is negligible.

A key component of our simulated dynamics is the non-exponential nature of the quintet population rise shown in Figs. 2a and 3. This is due to dynamics via the $^3$MC states, which is in line with the interpretation of ballistic dynamics observed experimentally[9]; note that without the $^3$MC, no such ballistic dynamics are observed, see Fig. 2b. The driving force for this non-exponential behaviour is the nuclear dynamics in the $^3$MC states, dominated by the impulsive expansion of the Fe-N bonds (see Fig. 4). These structural dynamics drive the molecule toward the intersection of the $^3$MC-$^5$MC PESs, at which efficient ISC occurs owing to direct $^3$MC-$^5$MC SOC. This also explains why the direct pathway is not operative in the presence of $^3$MC states: the nuclear dynamics on the steep $^3$MC surface promptly move the molecule away from the region where the $^3$MLCT→$^5$MC transition could be possible via large energetic overlap.

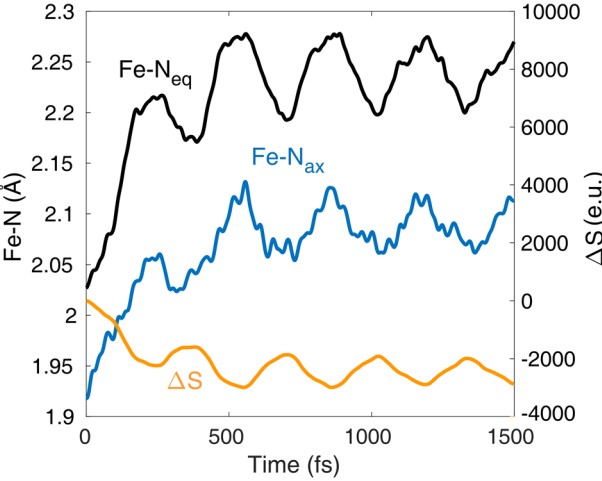

**Fig. 4 Average trajectories along Fe-N$_{ax}$ and Fe-N$_{eq}$ and the simulated difference X-ray scattering signal $\Delta S$.** N$_{ax}$ and N$_{eq}$ denote the axial and equatorial N atoms, see Fig. 1a. $\Delta S$ is referenced to $t = 0$ fs, at $q = 0.5$ Å$^{-1}$, which is characteristic for Fe(II) polypyridines.

The $^3$MC and $^5$MC populations exhibit oscillations with a ~300 fs period (which corresponds to the breathing mode), as seen in Fig. 2a. We find that this is a consequence of population transfer between $^3$T$_{1g}$, the lower $^3$MC component, and the $^5$MC states (see Supplementary Figure 2). Its origin can be identified by Fig. 4, which displays the time evolution of the average Fe-N bond lengths (black and blue curves). We assign these oscillations to the coherent nuclear dynamics along the breathing motion, which is activated in the MC states. In agreement with the time-resolved X-ray scattering experiment[9], we find that the structural Fe-N variations are slightly delayed with respect to the $^{1,3}$MLCT→$^3$MC electronic population dynamics. The reason for this is that only the MC states can launch the ballistic Fe-N expansion, which first have to be populated from the MLCT states. We interpret the large increase in Fe-N bond lengths in the MC states, as driving force of the vibrational coherence. Our coherent oscillations are consistent with various time-resolved experiments[5,7,9,24], however, in contrast to the experimental observations, we do not observe decoherence (damping). This is due to the absence of vibrational cooling (i.e., a solvent) in our theoretical model, which clearly cannot affect the low-spin→high-spin (quintet) process that is several times faster. The lack of the inclusion of damping is causing the only relevant difference from experiments, as in the latter only the first hump of the population oscillation is seen due to decoherence. Finally, we calculated the transient X-ray scattering data from the simulated evolution of the molecular structure. In agreement with a series of ultrafast studies on transition-metal complexes[9,25–27], we find that the coherent oscillations are directly observable by X-ray scattering, see the simulated difference signal displayed in yellow in Fig. 4.

## Conclusion

In the present work, we revealed the branching mechanism of Fe(II) polypyridine complexes by full-dimensional simulation of the entire singlet-triplet-quintet dynamics of [Fe(terpy)$_2$]$^{2+}$. We found that the quintet high-spin state is populated on the sub-ps timescale by two sequential pathways involving the two $^3$MC components $^3$T$_{1g}$ and $^3$T$_{2g}$. Importantly, we observe non-exponential population dynamics and coherent oscillations, which emphasises the essence of explicit dynamics frameworks. These results are consistent with various experimental time-resolved data, offer decision to a decade-long debate, and demonstrate the power of our theoretical dynamics approach to complement and interpret ultrafast experiments.

## Methods

Our dynamics methodology is based on full-dimensional trajectory surface hopping (TSH) in conjunction with a linear vibronic coupling (LVC) model[28,29]. We follow a hybrid approach[30], recently developed by one of us, based on the combination of time-dependent density functional theory (TD-DFT) PESs and multiconfigurational second-order perturbation theory (CASPT2) SOCs; in Supplementary Note 3 (see also Supplementary Tables 1 and 2, as well as Supplementary Figs. 6 and 7), we validate this hybrid methodology in terms of the correspondence of the DFT/TD-DFT vs CASPT2 states. Further methodological details are described below and under Supplementary Methods.

**Quantum chemistry.** The LVC potentials are based on B3LYP*/TZVP; the hybrid B3LYP* exchange-correlation functional was selected for its known accuracy for excited-state energetics of Fe(II) complexes[16,19,30–32]. Two-electron integrals were approximated by the resolution of identity (RI-J)[33] and chain-of-spheres (COSX)[34] methods. For TD-DFT caclulations, we used the Tamm-Dancoff approximation (TDA)[35]. The singlet and triplet excited states were calculated using singlet restricted Kohn-Sham (RKS) referenced TD-DFT, while for the quintet the reference was calculated by quintet unrestricted DFT; in Supplementary Note 4 (see also Supplementary Figures 8–11), by benchmarking against reference CASPT2 calculations, we show that the utilised DFT/TD-DFT methodology delivers reasonably accurate excited-state energetics. We note that solvent effects on the excitation energies are rather small, ~0.02 eV or below, as found at the FC geometry using a conductor-like polarisable continuum model (C-PCM)[36] for water. All DFT/TD-DFT calculations we carried out using the ORCA5.0[37,38] software.

The SOC matrix was calculated by complete active space self-consistent field (CASSCF) /CASPT2 calculations. We used an active space consisting of 10 electrons correlated on the following 16 orbitals: three 3$d$-t$_{2g}$ ($d_{xy}$, $d_{xz}$, $d_{yz}$) and two 3$d$-e$_g^*$ ($d_{x^2-y^2}$, $d_{z^2}$) type Fe-based orbitals, an additional set of three 4$d$-t$_{2g}$ and two 4$d$-e$_g^*$ orbitals, two Fe-N $\sigma$-bonding 3$d$-e$_g$ orbitals, and four dominantly ligand-based (terpy-$\pi^*$) orbitals that are required to access the MLCT states. We used atomic natural orbital relativistic correlation consistent (ANO-RCC) basis sets[39–41] with the following contractions: (7s6p5d4f3g2h) for Fe, (4s3p1d) for N, (3s2p) for C, and (2s) for H atoms. In the CASPT2 computations, we employed an imaginary level shift of 0.2 a.u. and a standard IPEA shift of 0.25 a.u. For CASPT2, we froze the core orbitals; in addition for CASSCF, we also froze the semi-core Fe-3$s$ and Fe-3$p_z$ orbitals, which were required to construct the 10e16o active space.

The CASSCF/CASPT2 calculations were performed at the CASPT2(10e,12o) ground-state ($^1$GS) minimum resulting from a 2D PES scan along the Fe-N$_{ax}$ bond length and the NNN angle[16]. We utilised C$_2$ point group symmetry with the symmetry axis $z$ defined by the Fe-N$_{ax}$ bonds. All CASSCF calculations were carried out with state averaging (equal weights) over 20 states (for each spin multiplicity singlet/triplet/quintet and C$_2$ state symmetry A/B) with the exception of triplet A, for which we had to use 30 roots to maintain the active space. The CASPT2 calculations were carried out in multistate (MS-CASPT2) mode using 10 states for each spin multiplicity/C$_2$ state symmetry. The SOC matrix elements were calculated by a spin-orbit state interaction (SO-SI) method[42,43] utilising MS-CASPT2 energies, CASSCF wave functions, and a one-electron effective mean-field SOC Hamiltonian[44]. Scalar relativistic effects were taken into account using the Douglas-Kroll-Hess (DKH) Hamiltonian[45,46]. All CASSCF/CASPT2 calculations were carried out using the OpenMolcas20.10[47,48] software.

**TSH dynamics.** The TSH methodology is based on Tully's fewest switches[49] a three-step propagator technique[50], and local diabatisation[51]. The simulations utilise the diagonal electronic basis, which is obtained by diagonalisation of the LVC potential matrix defined in Supplementary Equations 1 and 2 in Supplementary Methods. We used a time step of 0.5 fs and 0.005 fs for the nuclear and electronic propagation, respectively. 1000 initial conditions were sampled from a ground-state Wigner distribution, which were filtered using a stochastic algorithm accounting for conditions of the excitation process, i.e., excitation energies and oscillator strengths. We utilised a 0.1 eV wide energy window centred at 2.358 eV, which is the excitation energy of the lowest-lying optically-active $^1$MLCT state at the FC geometry. This process resulted in 716 selected initial conditions, from which we propagated the corresponding trajectories for 1.5 ps time duration. We used the energy-based method of Granucci et al.[52] to correct electronic decoherence effects utilising a decoherence parameter of 0.1 a.u. The number of trajectories (716) ensured convergence of the simulated excited-state dynamics. The difference X-ray scattering signal $\Delta S(t) = S(t) - S(0)$ was calculated by evaluating the Debye equation[53] for each structure and averaging over all trajectories. The TSH dynamics simulations were performed using the SHARC2.1 software[54–56].

## Data availability

Numerical parameters of the utilised model (Supplementary Data 1), as well as initial geometries, velocities (given in atomic units), and initial state indices corresponding to the adiabatic/spin-diabatic electronic basis (Supplementary Data 2) are provided in supplementary data files. Further data is available from the corresponding author upon reasonable request.

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

## Acknowledgements
The research leading to the presented results has received funding from the Hungarian National Research, Development and Innovation Fund under grant numbers NKFIH PD 134976 and 2018-1.2.1-NKP-2018-00012, and the Government of Hungary and the European Regional Development Fund under Grant No. VEKOP-2.3.2-16-2017-00015. M.P. acknowledges support from the János Bolyai Scholarship of the Hungarian Academy of Sciences. Guest access to the central HPC cluster of the Technical University of Denmark is acknowledged[57]. The authors are grateful to Dániel Buzsáki for his comments on the manuscript.

## Author contributions
G.V. conceived the project. M.P. developed the theoretical model. T.R. and M.P. carried out the simulations and analysed the obtained data. M.P. wrote the manuscript with critical feedback from all authors.

## Competing interests
The authors declare no competing interests.
