## [Peer Review File · Communications Chemistry]

Reviewers' comments:

Reviewer #1 (Remarks to the Author):

In this manuscript, Pápai and coworkers report on a theoretical study of the photodeactivation of $[\text{Fe}(\text{terpy})_2]^{2+}$, using trajectory surface hopping (TSH) on full-dimensional, linear vibronic coupling (LVC) model potentials. In particular, the authors focus on the cascade of intersystem crossing events in this molecule between singlet/triplet/quintet states – studied experimentally with numerous time-resolved techniques. The authors find that the nonradiative deactivation of $[\text{Fe}(\text{terpy})_2]^{2+}$ follows two distinct pathways after reaching triplet MLCT states, and that the triplet MC states play a decisive role in the deactivation of $[\text{Fe}(\text{terpy})_2]^{2+}$. The involvement of the triplet MC states further rationalizes the non-exponential growth of the quintet-state population.

This manuscript reports on an impressive set of calculations pushing the boundaries of conventional TSH simulations by including explicitly intersystem crossing processes between singlet, triplet, and quintet states for a sizable molecule with a large number of electronic states considered. The results are sound and explain experimental observations. They are further justified by the data proposed in the supplementary material and earlier work by the authors. I have a few comments below that the authors may want to consider, but I would definitely recommend this work for publication in Communications Chemistry.

1) My understanding is that the main experimental results used for comparison in this work (Ref. 9) were obtained in solvent (water). Considering the influence that water molecules may have on the energetics of MLCT states, I think that the authors should briefly discuss how accounting for solvent effects may alter the conclusion of this work (besides the lack of vibrational cooling already mentioned).

2) Out of curiosity, do the authors have validation data for the reliability of the LVC model when exploring the somewhat stretched Fe-N configurations close to what appears to be a minimum for the 5MC states? Can the MC states potentially lead to a single Fe-N bond rupture (with a possible bond reformation at later times, as postulated for other Ru/Ir polypyridyl-based complexes)?

As a final note: should the "Discussion" section perhaps be renamed "Conclusion"?

Reviewer #2 (Remarks to the Author):

In this manuscript, the authors simulate the full singlet-triplet-quintet dynamics of $[\text{Fe}(\text{terpyridine})_2]^{2+}$ by full dimensional spin-vibronic trajectory surface hopping and report a branching mechanism from 3MLCT component with two sequential components of 3MC, 3T1g and 3T2g. Furthermore, they observed that the 5MC is populated on the sub-picosecond timescale involving non-dimensional dynamics and coherent Fe-N breathing oscillations in agreement with the recent experimental results of a X-ray emission spectroscopy study on $[\text{Fe}(\text{bipyridine})_3]^{2+}$. In this perspective, this manuscript is very interesting since it allows responding to an experimental controversy. The originality of this theoretical study stems in the development of a dynamics methodology based on full-dimension trajectory surface hopping combined to a linear vibronic coupling (LVC) model. They adopt a hybrid approach recently developed by them that combines TD-DFT potential energy surfaces and multiconfigurational second order perturbation theory spin-orbit coupling calculations (CASPT2 SOC). The calculations are performed carefully and competently. The manuscript is well written according to a well-designed plan and the development has a logical build-up with clear to follow explanations of the work done and why. In this respect, I would like to

recommend it for publication on after clarifying the following issues.

- How in practice do the authors proceed to exclude the 3MC states in the simulations?
- Can the authors explain why in the simulations without the 3MC states (Figure 2)b), the population of the 1MC states increases again after 100 fs and by what mechanism?
- The LVC potential is based on the harmonic oscillator approximation and normal modes. How can the anharmonicity influence the dynamic of these states and is it possible to take it into account?

In more details:

- Caption of figure 1: Can you give a more detailed explanation of the two series of distances above the two figures: which states? and Why 5 Fe-Neq and 5 Fe-Nax ? Indeed, for me, there is four axial and two equatorial nitrogen positions.
- Can the authors define the acronym ISC
- I suggest to the authors to merge the Results and Discussion sections into one Results and Discussion section. Indeed, most of the discussion are in the Results section and the Discussion appears more like a conclusion.

Reviewer #3 (Remarks to the Author):

In their work, Rozgonyi et al. study the low-spin to high-spin photodynamics of a Fe(II) polypyridyl complex, $[\text{Fe}(\text{tpy})_3]^{2+}$, using the combination of trajectory surface hopping on model potentials parametrized in a linear vibronic coupling approach. Importantly, they manage to include singlet, triplet, and quintet states through a combination of TDDFT/UDFT/CASPT2 calculations which allows them to simulate the full dynamics. In this way, they are able to provide a clear picture of the relaxation mechanism of $[\text{Fe}(\text{tpy})_3]^{2+}$, especially regarding the involvement of 3MC states in the population of the high-spin 5MC states. I think this topic is interesting to a wide range of both theoretical and experimental photochemists and should be published in Communications Chemistry, however, only after some minor revisions.

1. The title and the discussions part claim insight for Fe(II) polypyridyl complexes (plural), however, since only one such complex is studied, the claim should only extend to one complex based on the present work.
2. The introduction cites experimental work regarding the singlet-to-quintet relaxation time for $[\text{Fe}(\text{bpy})_3]^{2+}$ (<200 fs). What about experimental work on $[\text{Fe}(\text{tpy})_3]^{2+}$?
3. The term "non-exponential dynamics" is used in several instances and I think it should be explained. It is brought into context by the "non-exponential rise of the quintet population". While the terms "exponential"/"non-exponential" appropriately describe the form of the curve of the quintet populations in the different simulations, I think a possibly better-suited term to describe the dynamics would be first-order or non-first order reactions (giving rise to exponential/non-exponential quintet curves).
4. The choice of using different shades of green to denote 3MLCT and different 3MC states makes their curves in all figures difficult to distinguish. The authors should use other color sets with more easily distinguishable colors.

5. Please add units to the normal mode coordinates in Figure 1.
6. The authors use diabatic electronic states in their discussion of the electron dynamics of the simulation. While this is explained in Supplementary Note 2, it should also be noted somewhere in the manuscript.
7. The diabatic states are labeled 1MC,3MC,1MLCT,3MLCT,5MC based on reference states at the FC geometry. How is the state composition of these states at the FC geometry, e.g., how much MLCT/MC character do the MLCT/MC states really possess? This could be answered through a transition-density-matrix analysis (JCP 152, 084109, 2020). Of course, such a discussion could be part of the SI.
8. The mechanisms of the dynamics simulations seem to be derived from visual inspection of the population curves. Especially for the distinguishment between the fast 3MLCT \rightarrow 3T2g \rightarrow 5MC and slow 3MLCT \rightarrow 3T2g \rightarrow 3T1g \rightarrow 5MC reaction, this derivation is difficult to follow, even for a trained eye. For example, from the shape of the curves, why could there be no direct 3MLCT \rightarrow 3T1g \rightarrow 5MC reaction? Thus, the suggested mechanisms should be supported by additional analysis. For this, the authors could use their proposed reaction schemes and fit kinetic models, where a close similarity between the fitted curves and the curves from the simulation would confirm their mechanism. Alternatively, the authors could analyze a representative amount of trajectories and follow the different diabatic states in time (Chem. Sci. 12, 10791, 2021).
9. The authors claim in Figure 2a very good overall agreement with the results of a study in reference 9. Please specify which results exactly are in good agreement. This could possibly be done in detail also in the SI.
10. Figures 2 and 3 show the excited-state population obtained from the dynamics and the caption of Figure 2 reports that the ground-state populations are excluded in this representation and discussed in the SI. However, I think this should also be mentioned in the text part, otherwise, it is very easy to miss and potentially misleading.
11. The authors assign the oscillations in Figure 4 in to the coherent nuclear dynamics along the breathing mode that is activated in the quintet state. Shouldn't the breathing mode be active in all MC states, only around different equilibrium values? This can be seen nicely in Figure 4, e.g., for the Fe-Nax distances which start at ca. 1.95 Å, then move to around 2.05 Å before ending up oscillating around 2.10-2.15 Å in the same time intervals as the transitions to the triplet and quintet manifolds occur.
12. The authors calculate the X-RAY scattering signal in Figure 4 and report good agreement with a series of experimental signals. Can the authors provide a comparison figure (e.g, in the SI) that supports this claim?
13. The authors used unrestricted DFT to calculate the energies of the quintet states while all other states were calculated using TDDFT based on a restricted DFT ground state. As these are different methods (see, e.g, the controversy discussed in JPCL 8, 5643, 2017), can the authors show that this combination can be justified? How well do the relative energies of the singlet/triplet vs quintet "excited" states compare? Should the quintet state possibly be shifted in energy relative to the singlet/triplet states?
14. The authors use CASPT2 to calculate SOCs and combine them with the potentials calculated at UDFT/TDDFT levels of theory. This combination is justified by using a diabatic representation for the

LVC potentials where all states are expressed in terms of the reference states at the FC geometry. However, isn't another necessary criterion also that the CASPT2 and UDFT/TDDFT states at the FC geometry are the same/similar? Can the authors demonstrate this (also quantitatively, e.g., through calculation of wave function overlaps)?

26 October 2022

Wigner Research Centre for Physics
P.O.B. 49, H-1525 Budapest, Hungary
Phone: +36-30-127-0088
Email: papai.matyas@wigner.hu

Dear Reviewers,

We are grateful for your careful evaluation of our manuscript

Branching mechanism of photoswitching in Fe(II) polypyridyl complexes explained by full singlet-triplet-quintet dynamics

submitted for publication in Communications Chemistry.

Please find below our responses, in which we address all your comments on our manuscript.

Reviewer #1:

1) My understanding is that the main experimental results used for comparison in this work (Ref. 9) were obtained in solvent (water). Considering the influence that water molecules may have on the energetics of MLCT states, I think that the authors should briefly discuss how accounting for solvent effects may alter the conclusion of this work (besides the lack of vibrational cooling already mentioned).

We have tested the effect of a solvent (water) on the TD-DFT excitation energies at the Franck-Condon (FC) geometry utilising a conductor-like polarizable continuum model (C-PCM). We observed a rather small solvent effect of 0.02 eV or below, even for the MLCT states. We introduced a note on this in the main text, under “Methods/Quantum chemistry”: “We note that solvent effects on the excitation energies are rather small, ~0.02 eV or below, as found at the FC geometry using a conductor-like polarisable continuum model (C-PCM)³⁶ for water.” (page 9).

2) Out of curiosity, do the authors have validation data for the reliability of the LVC model when exploring the somewhat stretched Fe-N configurations close to what appears to be a minimum for the 5MC states? Can the MC states potentially lead to a single Fe-N bond rupture (with a possible bond reformation at later times, as postulated for other Ru/Ir polypyridyl-based complexes)?

First, we would like to point out that the dominant dynamics leading to the population of the quintet (5MC) state are expected to occur relatively close to the FC geometry, where the LVC model is a good approximation. The 5MC minimum represents an extreme case where the LVC model is certainly less appropriate but this will not have any significant effect on how the 5MC is populated, which is the main focus of the present work. It will though affect the LS-HS(5MC) energy gap, but this is most relevant for the HS(5MC) \rightarrow LS relaxation, which takes place on the ns timescale and is out of the scope of this study.

We are not aware of any indication of photoinduced Fe-N bond rapture in $[\text{Fe}(\text{terpy})_2]^{2+}$ or $[\text{Fe}(\text{bipy})_3]^{2+}$, we note that especially the latter complex has been very extensively studied. Most likely, the reason is that the available vibrational energy is not sufficient that could lead to cleavage of a Fe-N bond.

As a final note: should the "Discussion" section perhaps be renamed "Conclusion"?

We agree with the reviewer and have changed the section title to "Conclusion".

Reviewer #2:

- How in practice do the authors proceed to exclude the 3MC states in the simulations?

We simply deleted each electronic parameter (epsilon, kappa, lambda, SOC) which involved a ^3MC state (T_1 - T_6) and re-indexed the triplet states. The number of spin-free triplet states was then reduced from 13 to 7 (all of them having $^3\text{MLCT}$ character).

- Can the authors explain why in the simulations without the 3MC states (Figure 2)b), the population of the 1MC states increases again after 100 fs and by what mechanism?

This can be explained using Fig. 1a, which shows the potential energy surfaces (PESs) along the dominant Fe-N breathing mode. In the simulation with the ^3MC states excluded, the ^1MC population increases due to the accessible $^1,^3\text{MLCT} / ^1\text{MC}$ crossings. This process is not observed for the full simulation because the ^3MC states efficiently drain the population away before it could be transferred to the ^1MC states (see the fast population increase of ^3MC states in Fig. 2a).

- The LVC potential is based on the harmonic oscillator approximation and normal modes. How can the anharmonicity influence the dynamic of these states and is it possible to take it into account?

Anharmonicity effects should be weak for the studied dynamics as no large-amplitude nuclear motion occurs. Even at extreme geometries, such as those with stretched Fe-N configurations around the ^5MC minimum ($\Delta R_{\text{Fe-N}} \sim 0.2 \text{ \AA}$), the harmonic approximation works rather well (see e.g., J. Chem Theory Comput. 2022 DOI: 10.1021/acs.jctc.1c01184). Anharmonicity is thus not significant for the studied dynamics. (It is possible to take anharmonicity into account by on-the-fly TSH simulations, but for $[\text{Fe}(\text{terpy})_2]^{2+}$, it would be computationally far too demanding and not even compatible with constant CASPT2 SOCs.)

In more details:

- Caption of figure 1: Can you give a more detailed explanation of the two series of distances above the two figures: which states? and Why 5 Fe-Neq and 5 Fe-Nax ? Indeed, for me, there is four axial and two equatorial nitrogen positions.

The two series of Fe-N distances refer to the two type nitrogen positions: axial ("ax", within the middle pyridine ring) and equatorial ("eq", within the outer pyridine rings). There are two axial and four equatorial nitrogen positions in the molecule. The five Fe-N_{ax} and Fe-N_{eq} values above Fig. 1a correspond to five different values of the dimensionless Fe-N breathing normal mode coordinate: -20, -10, 0, 10, and 20. Please note that respectively the two Fe-N_{ax} and four Fe-N_{eq} values are identical for a given value of the normal mode coordinate along this breathing mode, as it maintains the full symmetry of the molecule, i.e., the breathing mode is totally

symmetric. This is, however, not true for the antisymmetric mode shown in Fig. 1b which breaks the symmetry (there are thus two sets of both Fe-N_{ax} and Fe-N_{eq} values at -10 and 10 values along this normal mode).

- Can the authors define the acronym ISC

ISC stands for “intersystem crossing” and is defined at its first occurrence in the manuscript (page 2).

- I suggest to the authors to merge the Results and Discussion sections into one Results and Discussion section. Indeed, most of the discussion are in the Results section and the Discussion appears more like a conclusion.

We agree with this suggestion and have thus renamed the “Results” section to “Results and discussion” and changed the “Discussion” section title in the revised manuscript to “Conclusion”.

Reviewer #3:

1. The title and the discussions part claim insight for Fe(II) polypyridyl complexes (plural), however, since only one such complex is studied, the claim should only extend to one complex based on the present work.

Although it is correct that we only studied one complex, it is very likely that the proposed mechanism is general, also valid for [Fe(bipy)₃]²⁺. The potential energy landscape is very similar for the Fe(II) complexes with strong-field polypyridyl ligands, particularly close to the FC point, thus we believe that the dynamics of the first picosecond, when the system quickly passes through many states to reach the quintet state, are very similar, thus our model dynamics are likely to be generally applicable to them. So far, all available experimental evidence appears to support this claim. In fact, we contrasted our results to time-resolved experimental data to that of the [Fe(bipy)₃]²⁺ complex in order to assess the quality, as the latter is ample, already extensively analysed and readily available in the literature.

2. The introduction cites experimental work regarding the singlet-to-quintet relaxation time for [Fe(bpy)₃]²⁺ (<200 fs). What about experimental work on [Fe(tpy)₃]²⁺?

Transient optical absorption data shows that this timescale is very similar for [Fe(terpy)₂]²⁺, see e.g., A. Hauser *et al.*, *Coord. Chem. Rev.* 2006 DOI: 10.1016/j.ccr.2005.12.006. Our group has also obtained an extended set of experimental data on [Fe(terpy)₂]²⁺ (i.e., femtosecond-resolved transient optical absorption, X-ray emission spectroscopy and X-ray solution scattering), but they are not yet published as their analysis is not yet finalised. However, the preliminary results are close to being identical to those obtained on [Fe(bipy)₃]²⁺.

3. The term “non-exponential dynamics” is used in several instances and I think it should be explained. It is brought into context by the “non-exponential rise of the quintet population”. While the terms “exponential”/“non-exponential” appropriately describe the form of the curve of the quintet populations in the different simulations, I think a possibly better-suited term to describe the dynamics would be first-order or non-first order reactions (giving rise to exponential/non-exponential quintet curves).

“Non-exponential” in our manuscript refers to the population dynamics (i.e., the shape of the quintet population curve, as pointed out by the reviewer); wherever necessary, we made this explicitly clear in the text. We interpret and explain the observed non-exponential population dynamics as consequence of “nuclear dynamics in the ³MC states, dominated by the impulsive expansion of the Fe-N bonds“ (see the text in the revised manuscript, page 7, top). We do not think that discussion based on kinetic orders is well-suited for our case, since all the processes that lead to the quintet are first order in nature. However, in these relaxation processes, ballistic nuclear dynamics play an important role (which results in a non-exponential population change, as opposed to the case, in which the ³MC states are excluded, showing clear exponential kinetic behaviour, see Fig. 2b).

4. The choice of using different shades of green to denote 3MLCT and different 3MC states makes their curves in all figures difficult to distinguish. The authors should use other color sets with more easily distinguishable colors.

We made improvements such that the green colors are now distinguishable.

5. Please add units to the normal mode coordinates in Figure 1.

We use mass-frequency scaled normal coordinates, which are dimensionless. We added this information to the caption of Figure 1.

6. The authors use diabatic electronic states in their discussion of the electron dynamics of the simulation. While this is explained in Supplementary Note 2, it should also be noted somewhere in the manuscript.

We made this clear in the main text of the revised manuscript: “diabatic potentials” (caption of Fig. 1, page 3), “diabatic potential energy curves” and “diabatic excited-state populations” (page 4).

7. The diabatic states are labeled 1MC,3MC,1MLCT,3MLCT,5MC based on reference states at the FC geometry. How is the state composition of these states at the FC geometry, e.g., how much MLCT/MC character do the MLCT/MC states really possess? This could be answered through a transition-density-matrix analysis (JCP 152, 084109, 2020). Of course, such a discussion could be part of the SI.

The FC reference states have clearly dominant MLCT or MC character. In all cases, the weight of a mixing component is negligible with a maximum value of 10%, mostly, this MLCT/MC mixing weight is significantly smaller. We made this clear in the SI, Supplementary Note 2.

8. The mechanisms of the dynamics simulations seem to be derived from visual inspection of the population curves. Especially for the distinguishment between the fast 3MLCT → 3T_{2g} → 5MC and slow 3MLCT → 3T_{2g} → 3T_{1g} → 5MC reaction, this derivation is difficult to follow, even for a trained eye. For example, from the shape of the curves, why could there be no direct 3MLCT → 3T_{1g} → 5MC reaction? Thus, the suggested mechanisms should be supported by additional analysis. For this, the authors could use their proposed reaction schemes and fit kinetic models, where a close similarity between the fitted curves and the curves from the simulation would confirm their mechanism. Alternatively, the authors could analyze a representative amount of trajectories and follow the different diabatic states in time (Chem. Sci. 12, 10791, 2021).

We have investigated this by analysing the diabatic state populations ($^3\text{MLCT}$, $^3\text{T}_{1g}$, $^3\text{T}_{2g}$, ^5MC) along a representative set of 100 trajectories. We found that most trajectories follow the $^3\text{MLCT} \rightarrow ^3\text{T}_{2g} \rightarrow ^3\text{T}_{1g} \rightarrow ^5\text{MC}$ pathway, the $^3\text{MLCT} \rightarrow ^3\text{T}_{2g} \rightarrow ^5\text{MC}$ channel appears as a faster but minor component. We also found a single example (out of 100 analysed trajectories) for the $^3\text{MLCT} \rightarrow ^3\text{T}_{1g} \rightarrow ^5\text{MC}$ direct pathway mentioned by the reviewer, but its weight is so low that it is negligible. We have included this analysis in the SI in Supplementary Note 3, and given an example for trajectories following the three mechanisms leading to the population of the quintet state (Supplementary Figures 3–5), as well as changed the main text and abstract accordingly.

9. The authors claim in Figure 2a very good overall agreement with the results of a study in reference 9. Please specify which results exactly are in good agreement. This could possibly be done in detail also in the SI.

We made this specification in the main text of the revised manuscript: “decay of MLCT states, participation of ^3MC states, quintet population rise” (bottom of page 4).

10. Figures 2 and 3 show the excited-state population obtained from the dynamics and the caption of Figure 2 reports that the ground-state populations are excluded in this representation and discussed in the SI. However, I think this should also be mentioned in the text part, otherwise, it is very easy to miss and potentially misleading.

We have included a note addressing this in the main text of the revised manuscript: “We note that the ground-state population is negligible for the present analysis, and is thus not shown here but discussed in the Supplementary Information, see Supplementary Note 2 and Supplementary Figure 2.” (page 4).

11. The authors assign the oscillations in Figure 4 in to the coherent nuclear dynamics along the breathing mode that is activated in the quintet state. Shouldn't the breathing mode be active in all MC states, only around different equilibrium values? This can be seen nicely in Figure 4, e.g., for the Fe-Nax distances which start at ca. 1.95 Å, then move to around 2.05 Å before ending up oscillating around 2.10-2.15 Å in the same time intervals as the transitions to the triplet and quintet manifolds occur.

We agree with the reviewer and changed the text accordingly: “...activated in the MC states.” (page 7).

12. The authors calculate the X-RAY scattering signal in Figure 4 and report good agreement with a series of experimental signals. Can the authors provide a comparison figure (e.g, in the SI) that supports this claim?

What we state is not that our calculated X-ray scattering signal agrees well with the experimental ones, but that the observed “coherent oscillations are consistent with various TR experiments” and “coherent oscillations are directly observable by X-ray scattering” (page 8), both in our calculations and the quoted ultrafast experiments. We also mention that these time-resolved experiments were performed not on $[\text{Fe}(\text{terpy})_2]^{2+}$ but other transition-metal complexes. Nevertheless, comparing Fig. 5 of ref. 9 ($[\text{Fe}(\text{bipy})_3]^{2+}$) with Figs. 2a and 4 reveals very good qualitative agreement. Therefore, we believe that the provided references to the experimental works suffice.

13. The authors used unrestricted DFT to calculate the energies of the quintet states while all other states were calculated using TDDFT based on a restricted DFT ground state. As these are different methods (see, e.g. the controversy discussed in JPCL 8, 5643, 2017), can the authors show that this combination can be justified? How well do the relative energies of the singlet/triplet vs quintet “excited” states compare? Should the quintet state possibly be shifted in energy relative to the singlet/triplet states?

We have demonstrated the accuracy of unrestricted quintet DFT in combination with DFT/TD-DFT for a series of complexes with a $\text{Fe}^{\text{II}}\text{N}_6$ core, by benchmarking against high-level CASPT2 calculations (see J. Chem. Theory Comput. 2013 DOI: 10.1021/ct300932n and 2022 DOI: 10.1021/acs.jctc.1c01184). Nevertheless, we have directly investigated the effect of shifting the energy of the quintet states on the simulated population dynamics for a set of 33 trajectories by replacing the quintet DFT $\epsilon^{(a)}$ values (vertical excitation energies at the FC geometry) by the CASPT2 ones; this leads to 0.2 eV positive shift of the quintet energies with respect to those of the singlet/triplet states. We found that there is no significant effect on the electronic populations, confirming the robustness of our simulated dynamics.

14. The authors use CASPT2 to calculate SOCs and combine them with the potentials calculated at UDFT/TDDFT levels of theory. This combination is justified by using a diabatic representation for the LVC potentials where all states are expressed in terms of the reference states at the FC geometry. However, isn't another necessary criterion also that the CASPT2 and UDFT/TDDFT states at the FC geometry are the same/similar? Can the authors demonstrate this (also quantitatively, e.g., through calculation of wave function overlaps)?

We agree that the FC DFT/TD-DFT and CASPT2 states have to be similar, i.e., their electronic character should be consistent. As written, in the SI, Supplementary Note 4, “The character of the DFT/TD-DFT and CASPT2 electronic states was checked for consistency.” This was done by the analysis of dominant electronic configurations (now included in the SI, Supplementary Table 1). We also note that the active space for CASSCF/CASPT2 was chosen to include all orbitals involved in generation of the relevant excited states (the five Fe-3d orbitals and the lowest four ligand terpy- π^* orbitals, two A and the other two having B C_2 point group symmetry, which were found to agree very well for DFT and the state-averaged active orbitals, see Supplementary Figures 6 and 7). Of course, the states will not be fully identical, but this is not a requirement for the methodology to be operational (consistent electronic character suffices). The adequacy of the utilised methodology is confirmed by the good agreement between our simulated population dynamics and those extracted from time-resolved X-ray emission spectroscopy data (ref. 9 in the revised manuscript).

Additional changes: i) we deleted the graphical abstract to comply with the journal's format, ii) we have included a reference for using the SHARC nonadiabatic dynamics software (reference 53 in the revised manuscript), and iii) we provided the initial geometries, velocities, and electronic state indices as supplementary data (and added this information under “Data availability” in the revised manuscript).

Yours sincerely,

Mátyás Pápai

On the behalf of all authors

Reviewers' comments:

Reviewer #1 (Remarks to the Author):

The authors addressed most of my comments, and I would just have a (minor) question for them

Regarding my first question and the influence of water on MLCT states, the authors have calculated with a PCM the variation of the MLCT transition energies at the Franck-Condon geometry. This is only partly the issue in the context of water solvation for polypyridyl complexes, as it was reported for similar, ruthenium-based trisbipyridine complexes that water molecules could lead to a localization of the MLCT excitation on one bpy ligand upon excited-state relaxation (see work by Chergui and Tavernelli). Would the authors expect similar behavior with the terpyridine ligands of their Fe complex?

Reviewer #2 (Remarks to the Author):

The authors have taken into account most of the comments, answered to all of them and in a majority in a satisfactory way. They made the major corrections as well as additional analysis. Furthermore, they have rewritten some parts and completed others. All these revisions contribute to improve the manuscript and correct some misleading. In my opinion, this revised manuscript deserves to be published.

Reviewer #3 (Remarks to the Author):

I thank Rozgonyi et al. for addressing the comments of my previous review and I thank the authors for the additional calculations and analysis they performed during this effort. While most of the points of my previous review have been clarified, there are still some points that the authors should address again. Keeping the numbering from my previous review

1. The authors reply that they expect their results obtained for one complex to hold for the class of polypyridyl iron(II) complexes based on the similarity of the potential energy landscape. While I do agree that this is a likely situation, I think, describing this situation should be part of the discussion, conclusion, or outlook part of the paper but not stated in the title, as there is no evidence in the present work that the conclusions extend to other (similar) systems. For example the title could be changed simply to "Branching mechanism of photoswitching in an Fe(II) polypyridyl complex explained by full singlet-triplet-quintet dynamics"

2. It is not clear from the author's reply if there is already time-resolved experimental data on the early-time photodynamics of $[\text{Fe}(\text{tpy})_3]^{2+}$ available. The reference given in the authors reply is a review on the high-spin low-spin relaxation. I understand that the authors have obtained an unpublished set of experimental data on $[\text{Fe}(\text{tpy})_3]^{2+}$, however, as long as it is not published and can be inspected, I don't think the authors can already claim agreement with experimental data.

3. ...

4. Some of the colors are still difficult to distinguish. Perhaps instead of using different shades of green, the authors could consider using different colors to distinguish the curves.

5. ...

6. ...

7. ...

8. The authors have investigated the dynamics along a set of 100 trajectories and found that the $3\text{MLCT} \rightarrow 3\text{T}2\text{g} \rightarrow 5\text{MC}$ channel appears to be faster than the main pathway $3\text{MLCT} \rightarrow 3\text{T}2\text{g} \rightarrow 3\text{T}1\text{g}$

>5MC, while still being only a minor component. Can the authors explain, why the faster pathway only accounts for a minor contribution?

9. ...

10. ...

11. ...

12. ...

13. In response to my inquiry about the validity of combining RKS/TDDFT for singlets/triplets with UKS for quintets, the authors reference two previous work. However, only one of them deals with the present complex. In this work (JCTC 9, 509, 2013), potential energy scans of low-lying singlet/triplet/quintet states are given along a combined coordinate that connects the low-spin and high-spin minima. However, there, for the triplet states, yet another protocol was by calculating them from a triplet reference state, unlike in the present paper, where the triplet states are calculated from a singlet reference state. In order to show that the present computational protocol of combining RKS/TDDFT for singlets/triplets with UKS for quintets is valid (for the presently studied complex), the authors should use exactly this protocol for exactly this complex and provide either also potential energy scans or table of relative energies of all states at different geometries (e.g., the minima of the singlet, triplet and quintet) to compare to their CASPT2 reference energies. The authors further performed additional simulations where they used CASPT2 energies for the quintet states in place of the UKS energies, and they reported that no significant changes in the dynamics were observed. However, no results were shown to actually support this claim. Please, in future replies, add evidence whenever new claims are stated.

14. In response to my inquiry about the (also) quantitative comparability of UDFT/TDDFT and CASPT2 states, the authors have added a qualitative characterization of the states in Supplementary Table 1 together with Figures of the ligand orbitals involved in these states, indeed demonstrating a qualitative agreement. Furthermore, the authors have in previous work also showcased the validity of their approach for a different metal complex (JCTC 18, 1329, 2022). However, this metal complex only possessed low-lying MC states, while the present complex shows a mixture of low-lying MC and MLCT states. Since this is the first study of its kind on metal complexes where both of these states are present, I do think the authors should provide some quantitative analysis on how similar the UDFT/TDDFT and CASPT2 states actually are to prove the validity of their approach for the studied complex.

23 November 2022

Wigner Research Centre for Physics
P.O.B. 49, H-1525 Budapest, Hungary
Phone: +36-30-127-0088
Email: papai.matyas@wigner.hu

Dear Reviewers,

We are grateful for your careful evaluation of our manuscript

Branching mechanism of photoswitching in an Fe(II) polypyridyl complex explained by full singlet-triplet-quintet dynamics

submitted for publication in Communications Chemistry.

Please find below our responses, in which we address all your comments on our manuscript.

Reviewer #1:

Regarding my first question and the influence of water on MLCT states, the authors have calculated with a PCM the variation of the MLCT transition energies at the Franck-Condon geometry. This is only partly the issue in the context of water solvation for polypyridyl complexes, as it was reported for similar, ruthenium-based trisbipyridine complexes that water molecules could lead to a localization of the MLCT excitation on one bpy ligand upon excited-state relaxation (see work by Chergui and Tavernelli). Would the authors expect similar behavior with the terpyridine ligands of their Fe complex?

We do expect a similar behavior for $[\text{Fe}(\text{terpy})_2]^{2+}$, but its relevance should be significantly lower as the involved MLCT states are rather short lived (their decay occurs on a ~ 100 fs timescale).

Reviewer #3:

1. The authors reply that they expect their results obtained for one complex to hold for the class of polypyridyl iron(II) complexes based on the similarity of the potential energy landscape. While I do agree that this is a likely situation, I think, describing this situation should be part of the discussion, conclusion, or outlook part of the paper but not stated in the title, as there is no evidence in the present work that the conclusions extend to other (similar) systems. For example the title could be changed simply to “Branching mechanism of photoswitching in an Fe(II) polypyridyl complex explained by full singlet-triplet-quintet dynamics”

We accepted the reviewer's suggestion and have changed the title to "Branching mechanism of photoswitching in an Fe(II) polypyridyl complex explained by full singlet-triplet-quintet dynamics".

2. It is not clear from the author's reply if there is already time-resolved experimental data on the early-time photodynamics of $[\text{Fe}(\text{tpy})_3]^{2+}$ available. The reference given in the authors reply is a review on the high-spin low-spin relaxation. I understand that the authors have obtained an unpublished set of experimental data on $[\text{Fe}(\text{tpy})_3]^{2+}$, however, as long as it is not published and can be inspected, I don't think the authors can already claim agreement with experimental data.

In the supplementary information of the paper [Liu et al. Chem. Commun. 2013 DOI: 10.1039/C3CC43833C], it was reported that the quintet state of $[\text{Fe}(\text{terpy})_2]^{2+}$ is populated with a characteristic time of 145 fs, based on transient optical absorption data. Also, oscillations with a period of 400 ± 60 fs were observed. We changed the text in the Introduction on page 2 to "...which in turn leads to conversion into the quintet HS state in <200 fs; this timescale is very similar for $[\text{Fe}(\text{terpy})_2]^{2+}$ (terpy = 2,2':6',2''-terpyridine), another important member of the polypyridine family, as observed by TOAS[liu2013]}."

4. Some of the colors are still difficult to distinguish. Perhaps instead of using different shades of green, the authors could consider using different colors to distinguish the curves.

Again, we accepted the reviewer's suggestion and have changed the color for the $^3\text{MC}(^3\text{T}_{2g})$ state to gold (and changed the notation for the ^1GS population to black dashed, for which we used gold in the earlier submissions).

8. The authors have investigated the dynamics along a set of 100 trajectories and found that the $3\text{MLCT} \rightarrow 3\text{T}_{2g} \rightarrow 5\text{MC}$ channel appears to be faster than the main pathway $3\text{MLCT} \rightarrow 3\text{T}_{2g} \rightarrow 3\text{T}_{1g} \rightarrow 5\text{MC}$, while still being only a minor component. Can the authors explain, why the faster pathway only accounts for a minor contribution?

A faster pathway can have a minor contribution. Our interpretation for this faster but minor component is that a smaller nuclear gradient drives this pathway, while the dominant relaxation channel is likely driven by a larger nuclear gradient. Each trajectory is started from an initial geometry and velocities; the present result simply means that there are fewer initial conditions leading to the fast relaxation channel than to the slower component.

13. In response to my inquiry about the validity of combining RKS/TDDFT for singlets/triplets with UKS for quintets, the authors reference two previous work. However, only one of them deals with the present complex. In this work (JCTC 9, 509, 2013), potential energy scans of low-lying singlet/triplet/quintet states are given along a combined coordinate that connects the low-spin and high-spin minima. However, there, for the triplet states, yet another protocol was by calculating them from a triplet reference state, unlike in the present paper, where the triplet states are calculated from a singlet reference state. In order to show that the present computational protocol of combining RKS/TDDFT for singlets/triplets with UKS for quintets is valid (for the presently studied complex), the authors should use exactly this protocol for exactly this complex and provide either also potential energy scans or table of relative energies of all states at different geometries (e.g., the minima of the singlet, triplet and quintet) to

compare to their CASPT2 reference energies. The authors further performed additional simulations where they used CASPT2 energies for the quintet states in place of the UKS energies, and they reported that no significant changes in the dynamics were observed. However, no results were shown to actually support this claim. Please, in future replies, add evidence whenever new claims are stated.

Regarding the combination of RKS/TD-DFT for singlets-triplets and UKS for quintets, using published computational data and previous test calculations, we below demonstrate that it reproduces rather well reference CASPT2 energetics. In the first figure, we present the comparison we mentioned in our previous reply from our JCTC paper [DOI: 10.1021/ct300932n] for the potential energy surfaces (PESs) of $[\text{Fe}(\text{terpy})_2]^{2+}$, but now with the same protocol utilised in the present work (RKS/TD-DFT singlet-triplet + UKS quintet) benchmarked against CASPT2. As is clear from the figure, these results demonstrate a reasonably good qualitative agreement, with one of the most significant differences indeed being the quintet overstabilisation by UKS with respect to CASPT2, as suggested by the reviewer in their original review reports.

Furthermore, we evaluated the analogous comparison for the closely-related $[\text{Fe}(\text{bipy})_3]^{2+}$ complex, for which PESs including those for MLCT states are available at the reference CASPT2 level (Sousa et al. Chem. Eur. J. 2013 DOI: 10.1002/chem.201302992). Here, as can be seen in the below figure, the same conclusions are reached as for $[\text{Fe}(\text{terpy})_2]^{2+}$: the overall agreement is rather good (note that the smaller number of curves is due to the higher symmetry of $[\text{Fe}(\text{bipy})_3]^{2+}$, the structure of $[\text{Fe}(\text{terpy})_2]^{2+}$ significantly departs from octahedral symmetry by axial distortion).

Finally, the third set of PESs demonstrates the close similarity of the excited-state energetics of $[\text{Fe}(\text{terpy})_2]^{2+}$ and $[\text{Fe}(\text{bipy})_3]^{2+}$; for both, we utilised the DFT/TD-DFT methodology used in the present work (RKS TD-DFT + UKS). This justifies the utilisation of the second set of PESs for $[\text{Fe}(\text{bipy})_3]^{2+}$, which is though not exactly the same complex as $[\text{Fe}(\text{terpy})_2]^{2+}$, but for benchmarking purposes the differences are clearly negligible.

We now present the result of our additional dynamics simulations, in which we shifted the energy of the quintet states by +0.2 eV, according to the CASPT2 energetics. We calculated 100 trajectories, the results are compared in the figure below to the population dynamics of our original simulation with unshifted energies. This figure demonstrates good overall agreement with the only notable difference in the timescale of the quintet population growth, which is faster for the new simulation, as expected from the reduction of the decisive triplet-quintet energy gaps around the FC geometry, caused by the +0.2 eV quintet energy shift. Importantly, all these results point in the same direction validating the accuracy of our DFT/TD-DFT methodology.

14. In response to my inquiry about the (also) quantitative comparability of UDFT/TDDFT and CASPT2 states, the authors have added a qualitative characterization of the states in Supplementary Table 1 together with Figures of the ligand orbitals involved in these states, indeed demonstrating a qualitative agreement. Furthermore, the authors have in previous work also showcased the validity of their approach for a different metal complex (JCTC 18, 1329, 2022). However, this metal complex only possessed low-lying MC states, while the present complex shows a mixture of low-lying MC and MLCT states. Since this is the first study of its kind on metal complexes where both of these states are present, I do think the authors should provide some quantitative analysis on how similar the UDFT/TDDFT and CASPT2 states actually are to prove the validity of their approach for the studied complex.

First, we would like to mention that we did confirm the *similarity* of the states, requested by the reviewer in their original review reports; the corresponding qualitative analysis is given in the SI. The states will certainly not be *identical*, as they are calculated at different levels of quantum chemistry.

Nevertheless, we carried out further quantitative analysis. We here note that quantitative analysis to assess the similarity of the states by wave function overlaps suggested by the reviewer in our case would be simply misleading, as the two geometries at which we carried out the DFT and the CASPT2 calculations are different; this is because the CASPT2 minimum is significantly shifted from the DFT one towards shorter Fe-N bond lengths, see: J. Chem. Theory Comput. 2013 DOI: 10.1021/ct300932n, as well as the first two figures of PESs presented in the response to point 13 (but surely other coordinates are also affected to some extent). Therefore, we do not see how overlaps over the two different geometries could be calculated in a meaningful way. However, we decided to carry out a quantitative analysis of spin-orbit couplings (SOCs), whose validity was in the focus of the original question. These are the only parameters of our model in which the CASPT2 states enter, and for which the geometry dependence is assumed to be small and neglected anyway, in the LVC models. For singlet-triplet and triplet-triplet SOC, we compare the CASPT2 and TD-DFT SOC calculated at the ground-state equilibrium geometry. Triplet-quintet SOC cannot be calculated within our DFT/TD-DFT approach, as RKS TD-DFT triplets are not compatible with UKS quintets. For these triplet-quintet SOC, we compare our CASPT2 SOC for $[\text{Fe}(\text{terpy})_2]^{2+}$ to the CASPT2 SOC for $[\text{Fe}(\text{bipy})_3]^{2+}$ (taken from Sousa et al. Chem. Eur. J. 2013 DOI:

10.1002/chem.201302992); this work also reports singlet-triplet SOC values which we also use in our comparative analysis. In the table below, the largest two SOC values are presented in cm^{-1} - with the real and imaginary parts combined into the absolute value - for each multiplicity and character. (For $[\text{Fe}(\text{bipy})_3]^{2+}$ the largest value is given, as only this was reported in the 2013 Chem. Eur J. paper. Furthermore, SOC values between triplet states were also not reported in the 2013 Chem. Eur J. paper).

SOC term	$[\text{Fe}(\text{terpy})_2]^{2+}$ - CASPT2	$[\text{Fe}(\text{terpy})_2]^{2+}$ - TD-DFT	$[\text{Fe}(\text{bipy})_3]^{2+}$ - CASPT2
$ \langle {}^1\text{MC} H_{\text{SOC}} {}^3\text{MC}({}^3\text{T}_{1g})\rangle $	130.2, 67.4	122.2, 71.7	75.5
$ \langle {}^1\text{MC} H_{\text{SOC}} {}^3\text{MC}({}^3\text{T}_{2g})\rangle $	163.4, 137.3	156.1, 115.4	131.4
$ \langle {}^1\text{MC} H_{\text{SOC}} {}^3\text{MLCT}\rangle $	178.2, 72.6	90.5, 73.8	164.7
$ \langle {}^1\text{MLCT} H_{\text{SOC}} {}^3\text{MC}({}^3\text{T}_{1g})\rangle $	100.5, 32.0	75.7, 48.7	96.0
$ \langle {}^1\text{MLCT} H_{\text{SOC}} {}^3\text{MC}({}^3\text{T}_{2g})\rangle $	149.9, 145.4	86.4, 57.4	214.3
$ \langle {}^1\text{MLCT} H_{\text{SOC}} {}^3\text{MLCT}\rangle $	175.8, 175.0	180.6, 170.5	199.9
$ \langle {}^3\text{MLCT} H_{\text{SOC}} {}^3\text{MC}({}^3\text{T}_{1g})\rangle $	83.3, 51.5	46.7, 42.6	–
$ \langle {}^3\text{MLCT} H_{\text{SOC}} {}^3\text{MC}({}^3\text{T}_{2g})\rangle $	119.3, 104.6	63.8, 51.8	–
$ \langle {}^3\text{MC}({}^3\text{T}_{1g}) H_{\text{SOC}} {}^3\text{MC}({}^3\text{T}_{2g})\rangle $	170.9, 93.5	71.2, 41.0	–
$ \langle {}^3\text{MLCT} H_{\text{SOC}} {}^5\text{MC}\rangle $	18.3, 12.5	–	6.2
$ \langle {}^3\text{MC}({}^3\text{T}_{1g}) H_{\text{SOC}} {}^5\text{MC}\rangle $	373.1, 355.5	–	417.7
$ \langle {}^3\text{MC}({}^3\text{T}_{2g}) H_{\text{SOC}} {}^5\text{MC}\rangle $	263.8, 262.7	–	219.9

While we see some systematic underestimation of the SOC values by the TD-DFT compared to the CASPT2 values, the agreement between the respective values is satisfactory and thus justifies the combination of DFT/TD-DFT PESs and CASPT2 SOC values for the studied $[\text{Fe}(\text{terpy})_2]^{2+}$ complex. In addition, when we compare the SOC values obtained by the same CASPT2 method for the two polypyridine complexes, $[\text{Fe}(\text{terpy})_2]^{2+}$ and $[\text{Fe}(\text{bipy})_3]^{2+}$, we also find a good agreement, which is consistent with the close similarity in the photophysical behaviour of the two molecules.

Yours sincerely,

Mátyás Pápai

On the behalf of all authors

REVIEWERS' COMMENTS:

Reviewer #3 (Remarks to the Author):

I thank Rozgonyi et al. for addressing the comments of my previous review and the additional analysis presented in their reply letter. All my questions have been fully answered by the authors. I would only like to ask the authors to include their analysis regarding question 13 (combination of RKS/TDDFT for singlets and triplet and UKS for quintets) and question 14 (influence of state character of TDDFT and CASPT2 states on SOCS) in their Supplementary Information so that it is documented and will also be available for other interested readers.

Overall, the authors have compiled an interesting study for experimentalists and theoreticians, giving valuable insight into the excited-state dynamics of an iron(II) polypyridyl complex as well as showcasing the successful combination of different levels of electronic structure theories for their dynamics simulation, which both will surely inspire future works. Thus, the manuscript will make an excellent contribution to Communications Chemistry and should be published.

30 November 2022

Wigner Research Centre for Physics
P.O.B. 49, H-1525 Budapest, Hungary
Phone: +36-30-127-0088
Email: papai.matyas@wigner.hu

Dear Reviewers,

We are grateful for your careful evaluation of our manuscript

Branching mechanism of photoswitching in an Fe(II) polypyridyl complex explained by full singlet-triplet-quintet dynamics

submitted for publication in Communications Chemistry.

Please find below our responses, in which we address all your comments on our manuscript.

Reviewer #3:

I thank Rozgonyi et al. for addressing the comments of my previous review and the additional analysis presented in their reply letter. All my questions have been fully answered by the authors. I would only like to ask the authors to include their analysis regarding question 13 (combination of RKS/TDDFT for singlets and triplet and UKS for quintets) and question 14 (influence of state character of TDDFT and CASPT2 states on SOCS) in their Supplementary Information so that it is documented and will also be available for other interested readers.

We have included these analyses in the Supplementary Information, Supplementary Notes 3 and 4.

Yours sincerely,

Mátyás Pápai

On the behalf of all authors